# Clinical Features of Infectious Uveitis in Children Referred to a Hospital-Based Eye Clinic in Italy

**DOI:** 10.3390/medicina58111673

**Published:** 2022-11-18

**Authors:** Maria Pia Paroli, Lucia Restivo, Eleonora Ottaviani, Chiara Nardella, Irene Abicca, Luca Spadea, Marino Paroli

**Affiliations:** 1Eye Clinic, Department of Sense Organs, Sapienza University of Rome, 00161 Rome, Italy; 2Post Graduate School of Public Health, University of Siena, 53100 Siena, Italy; 3Clinical Immunology Service, Department of Clinical, Cardiovascular and Anesthesiology Sciences, Sapienza University of Rome, 00185 Rome, Italy

**Keywords:** infectious uveitis, pediatric ophthalmology, toxoplasmosis, toxocariasis, viral infections

## Abstract

*Background and Objectives*: To investigate the etiology, clinical features, ocular complications, and visual outcomes in children with infectious uveitis referred to a tertiary uveitis hospital-based service. *Materials and Methods*: Children with infectious uveitis were included in a retrospective cohort study. The data set was obtained after reviewing the medical records of pediatric patients with uveitis of different causes referred to our center during the period from 2009 to 2019. Clinical evaluations were performed at the time of diagnosis and the end of follow-up. *Results*: Uveitis of infectious origin was present in 57 (72 eyes) of 314 (18.1%) patients examined. The median age at presentation was 10.9 years (6.1–15.8), 52.6% of patients were female, and 47.4% were male. The main cause of infectious uveitis was viral (56.1% of cases), followed by Toxoplasma gondii infection (24.5%). The anatomical location of uveitis was posterior in 40.3%, anterior in 36.8%, panuveitis in 15.7%, and intermediate in 7% of cases. Ocular involvement was unilateral in 42 children (73.7%) and bilateral in 15 (26.3%) cases. The main causes of reduced visual acuity were cataract and maculopathy in 57.1% and 28.5% of cases, respectively. During the follow-up period, 75% of patients showed significant improvements in visual acuity. *Conclusions*: Specialist management in a tertiary referral eye care center facilitates early diagnosis and effective treatment of this serious cause of morbidity and vision loss in children.

## 1. Introduction

Pediatric uveitis accounts for about 5–10% of total uveitis. It is a cause of significant morbidity that increases with the duration of the disease. Diagnosis in children is generally delayed, thus having a worse prognosis than adult uveitis [1]. Most studies on uveitis in children have focused on the non-infectious form. However, infectious uveitis often has a poor prognosis because of its high tendency to become chronic [2,3]. The most frequent ocular complications consist of the development of cataracts, band keratopathy, glaucoma, amblyopia, and macular disease [3]. All of these conditions can lead to reduced visual acuity and even blindness. In many studies, an infectious cause of uveitis has been identified in 6–33% of children seen in tertiary referral centers [4,5,6]. As in adults, the prevalence of infectious uveitis varies from country to country depending on migration flows and particular socioeconomic conditions. Most cases of infectious uveitis involve populations in the Middle East, Turkey, and Africa [1]. In Italy, an infectious origin has been found in one-third of cases of uveitis in the pediatric population [6]. Regarding the prevalence of the different etiologies, toxoplasmosis appears to be the most frequent cause of infectious uveitis worldwide, followed by forms with viral etiology and Toxocara canis [4]. In developing countries, tuberculosis is a significant cause of childhood uveitis, while post-streptococcal uveitis is mainly reported in Europe [7]. The difficulty and delay in diagnosis, limited treatment options, and the effect of the disease throughout adult life are major challenges for pediatric ophthalmologists. Therefore, the ability to identify infectious uveitis early is extremely important, as treatment regimens differ significantly between infectious and noninfectious uveitis. The purpose of this retrospective cohort study was to analyze the etiology, clinical features, ocular complications, and visual prognosis of a series of children with infectious uveitis referred to our tertiary eye care center and to compare the results obtained with those available in the literature.

## 2. Materials and Methods

### 2.1. Patient Population

A total of 57 patients under the age of 16 years with infectious uveitis were enrolled in this study from January 2009 to December 2019. Patient data were obtained from a total of 314 medical records of children with uveitis from various origins referred to the Uveitis Center of the Department of Sense Organs at the Sapienza University of Rome. The research was conducted according to the ethical principles of the Declaration of Helsinki. The study was approved by the Ethics Committee of our institution, and all patients provided written consent to the research. The patient population included 30 females and 27 males. All patients underwent a thorough history focusing on the age of onset of uveitis, duration of disease, and characteristics of disease onset.

### 2.2. Uveitis Clinical Evaluation

The anatomical location of the uveitis (anterior, intermediate, posterior, or panuveitis), the course of the disease (acute, chronic, or recurrent), the given treatment, any ocular complications, and the visual prognosis were also evaluated. Initial visual acuity was compared with that measured at the end of the follow-up period. Uveitis was classified according to criteria established by the International Uveitis Study Group [8]. Each patient underwent a comprehensive ophthalmologic examination that included a best corrected visual acuity test (BCVA) using Snellen charts or illiterate E charts, biomicroscopy with slit-lamp ocular evaluation, tonometry, and fundus oculi examination. BCVA was measured using a 20-foot visual acuity scale. Visual impairment was defined as 20/50 or worse, and legal blindness 20/200 or worse. The median follow-up period was 23.9 months (14.3–34.2).

### 2.3. Laboratory Investigation

A diagnosis of infectious uveitis was based on a combination of clinical and laboratory findings. Based on the diagnostic orientation and clinical picture of uveitis, patients underwent investigations for ocular infectious diseases, such as titration of virus-specific antibodies against Epstein–Barr virus (EBV), herpes simplex virus (HSV), varicella-zoster virus (VZV), cytomegalovirus (CMV), rubella virus (RBV), anti-streptolysin O test, serologic test for Toxoplasma gondii and Toxocara canis, Mantoux test, and chest X-ray. In doubtful cases, blood and aqueous humor samples were taken for the qualitative detection of pathogen DNA by polymerase chain reaction (PCR) performed at the Department of Infectious Diseases of the Policlinico Umberto I University Hospital. Table 1 summarizes the diagnostic tests used for the various infection.

### 2.4. OCT and Tonometry

The diagnosis of macular edema was based on clinical examination and optical coherence tomography (OCT). Ocular hypertension was defined by the presence of intraocular pressure values above 21 mmHg, ophthalmoscopic evidence of optic nerve head damage, and the presence of typical visual defects.

### 2.5. Statistical Analysis

Statistical analysis was performed using the GraphPad Prism 8.0.0 software for Windows, San Diego, CA, USA. Chi-square (χ^2^) test for categorical data was used. Differences between groups were considered significant when the *p* value was ≤ 0.05. Numerical values were presented as percentages or as medians and interquartile ranges (Q1–Q3).

## 3. Results

### 3.1. Group Characteristics

The diagnosis of infectious uveitis was made in 57 (72 eyes) out of 314 cases of pediatric uveitis due to all causes (18.1%). A total of 30 children were female (52.6%) and 27 were male (47.4%), the median age at diagnosis was 10.9 years (6.1–15.8). The median follow-up period was 23.9 months (14.3–34.2).

### 3.2. Etiology of Infectious Uveitis

Blood and aqueous humor samples had to be taken for the qualitative detection of pathogen DNA by polymerase chain reaction (PCR) in one child with acquired chorioretinal toxoplasmosis and aqueous humor in five children with CMV uveitis. In the other cases, a clinical diagnosis was made. Toxoplasma gondii was the agent responsible for 24.5% of cases. In most cases, toxoplasmosis was diagnosed clinically based on the presence of typical focal necrotizing retinochoroiditis. The diagnosis of congenital toxoplasmosis was possible in only three cases due to the detection of seroconversion in the mother in the second trimester of pregnancy. For the remaining cases, ocular toxoplasmosis was presumably acquired. Retinochoroiditis was predominantly unilateral, involving the posterior pole with macula impairment in six eyes. General history revealed the presence of risk factors in seven children. In particular, two ate raw meat and five had a cat with whom they had close and regular contact. Viral infections were responsible for 56.14% of the cases when considered as a whole. However, in this study, each virus was responsible for a relatively small number of cases. Fuchs’ heterochromic uveitis was diagnosed in five patients (8.7%). Streptococcal and Mycobacterium tuberculosis infections both caused uveitis in 3.5% of cases. The two children with ocular tuberculosis also had systemic disease with pulmonary involvement. Toxocariasis was found in the same percentage of patients. (Table 2).

### 3.3. Anatomical Location of Uveitis

Regarding the anatomical location of uveitis, the posterior form was the most common being found in 40.3% of cases, the anterior form in 36.8%, panuveitis in 15.7%, and the intermediate form in 7% of cases. Posterior uveitis was present in the cases of Toxoplasma gondii infection (60.9%) followed in percentage by CMV (21.7%) and Toxocara canis (8.8%). EBV and Streptococcus were both responsible for only 4.3% of the posterior uveitis observed in this study. Anterior uveitis was caused by HSV infection in 42.9% of cases, Fuchs’ heterochromic uveitis in 23.8%, VZV in 19%, CMV in 9.5%, and EBV in 4.8% of cases. Intermediate uveitis was caused by EBV infection in three cases and by CMV in one case. Panuveitis was observed in 33.4% of cases due to VZV infection, while 22.2% of cases were due to Mycobacterium tuberculosis or HSV infection. CMV and Streptococcus were responsible for 11.1% of cases (Table 3). Ocular involvement was bilateral in 26.3% of cases and unilateral in the remaining 73.7%. Bilateral forms were found in 19.4% of anterior uveitis, 25% of intermediate uveitis, 21.7% of posterior uveitis, and 55.5% of panuveitis.

### 3.4. Ocular Complications

Papillitis was the most frequent ocular complication, occurring in 17/72 eyes (23.6% of cases), followed by cataracts and posterior synechiae, present in 14/72 eyes (19.4% of cases). Papillitis was caused by VZV in four cases, Toxoplasma gondii in two cases, CMV in two cases, and EBV in a single case as well as the streptococcal infection. Cataract was observed in ten children, three with Fuchs’ heterochromic uveitis, two with uveitis associated with VZV or Mycobacterium tuberculosis infection, and three with CMV, EBV, and HSV infection, respectively. Posterior synechiae affected nine patients for a total of 14/72 eyes (19.4%). Of these patients, three had uveitis caused by HSV, two by VZV, and two by Mycobacterium tuberculosis; one case had Toxoplasmosis, and one had CMV infection. Typical fine keratic precipitate was found in all anterior viral uveitis, iris atrophy was found in half of the HSV cases, and elevated IOP was detected during an episode of flare in one eye with HSV and four eyes with VZV. Cystoid macular edema complicated uveitis in nine eyes of five cases (12.5%) including two due to Toxoplasma gondii infection and the remaining cases were due to EBV, VZV, and streptococcal infection. The presence of epiretinal membrane was present in six eyes of five cases (8.3%), caused by Toxocara canis, Toxoplasma gondii, EBV, VZV, and Mycobacterium tuberculosis, respectively.

### 3.5. Visual Acuity Assessment

The visual prognosis was assessed by measuring the changes in visual acuity (VA) at the end of the follow-up compared with the baseline values. At the end of the study, 77.8% of patients had visual acuity > 20/50 while in 22.2% it was <20/50. In 75% of cases, the visual prognosis was considered to be favorable with significant improvement in VA compared to the values measured at the time of the first visit. The most frequent cause of reduced visual acuity at the end of the follow-up period was cataracts, present in 57.14% of eyes with visual acuity < 20/50, followed by cystoid macular edema/maculopathy that was present in 28.57% of cases. The course of infectious uveitis in the children analyzed in this study showed that recurrent uveitis was the most frequent form, being present in 47.4% of cases, followed by chronic forms (38.6%) and acute forms (14%). Acute inflammation with a rapid course was found in 19.1% of anterior uveitis and 8.7% of posterior uveitis. Recurrent disease activity was detected in 57.1% of anterior uveitis, 47.8% of posterior uveitis, 33.3% of panuveitis, and 25% of intermediate uveitis. Otherwise, a chronic course was found in 55.6% of panuveitis, 50% of intermediate uveitis, 43.5% of posterior uveitis, and 23.8% of anterior uveitis. The median follow-up period was 18 months (11.8–23.5) for anterior uveitis, 39.5 months (26.2–52.4) for intermediate uveitis, 8.52 months (5.6–13.1) for posterior uveitis, and 32.3 months (21.2–46.7) for panuveitis. These data are summarized in Table 4. As shown in Table 5, the ocular symptoms at presentation were reduced visual acuity in 40.35% of cases, ocular redness in 36.84%, ocular pain in 24.56%, floaters in 17.54%, and photophobia in 14.04% of cases.

Reduced visual acuity (BCVA with loss of vision of one or more lines in Snellen tables or illiterate E tables) was found in all patients with posterior uveitis, regardless of etiology, while viral anterior uveitis was related to hyperemia and photophobia and in a single case of intermediate viral uveitis. However, it must be considered that multiple symptoms were present simultaneously in the same patient. Strabismus was detected in 17.4% of posterior uveitis cases and in 4.8% of anterior uveitis. Uveitis was treated according to international treatment guidelines.

## 4. Discussion

Infectious uveitis in pediatrics is rare compared with in adults. In a recent Italian study on uveitis of all causes, it was reported that 30% of cases were infectious in origin, but only 12% of these were in childhood [2]. The results obtained from the present study of a cohort of Italian children showed that 18.5% of uveitis from all causes was infectious in origin and illustrates a decrease in prevalence compared to 31% as reported in a previous study in the same tertiary referral center [6]. These findings are similar with those reported by Hettinga et al. [1]. The median age at diagnosis was 10.9 years (6.1–15.8). The incidence of infectious uveitis did not differ between males and females. In our study, ocular involvement was predominantly unilateral posterior forms of uveitis, constituting the largest group of cases (40.3%), due to the high frequency of toxoplasmosis. The second most common form was anterior uveitis (37%), which was predominantly associated with a viral etiology, caused in particular by HSV infection. A viral etiology caused predominantly by CMV (25%) and EBV (75%) was also found in intermediate uveitis and posterior uveitis (11.1%). Gautam et al. conducted a study in northern India, in which panuveitis was present in 16% of cases, of which 66.7% were related to viral agents [9]. Viral etiology has often been associated with anterior uveitis, but its frequency in panuveitis is due to the typical chronic/recurrent course of viral infections causing the extension of inflammation to posterior ocular components [10] often observed in childhood. The two cases of ocular tuberculosis described in our study had panuveitis. This is in agreement with Basu et al. [11], who reported that chronic diffuse uveitis and retinal vasculitis are the main ocular manifestations of this infection. Viral etiology was the main cause of uveitis in our children, present in 56.14% of cases, followed by Toxoplasma gondii infection, present in 24.56% of cases. In our study, the virus that most frequently caused uveitis was HSV, present in 15.79% of cases. In our study CMV infection was detected in 14.04% of cases, VZV infection in 28.12%, and EBV infection in 8.77% of cases. In contrast, BenEzra et al. found EBV as a cause of uveitis in 7.6% of cases, followed by CMV in 5.4% and VZV in 2.2%. Several studies point out that anterior uveitis is predominantly associated with a viral etiology. The viral agent most commonly involved in the genesis of anterior uveitis is the herpes simplex virus (HSV). HSV anterior uveitis generally affects patients around their fifth decade of life and tends to occur simultaneously with the reactivation of the virus in another district [12]. In about 20% of cases, herpes virus simplex uveitis turns out to be unilateral and may complicate with keratitis in about 40% of cases. Corneal manifestations can be either acute or chronic [13]. Uveitis caused by ZVZ is found to be present in about 50% of patients with ophthalmic herpes zoster [14]. Uveitis present during VZV infection is usually severe, as it is often associated with occlusive vasculitis. It has also been reported that iris hypotrophy associated with ZVZ infection is directly associated with the viral load of the virus within the aqueous humor [15]. ZVZ uveitis can also involve the cornea and sclera [16] as well as be responsible for secondary glaucoma [17]. All these findings underline the importance of posterior segment analysis in all cases of uveitis from herpetic viruses. Another important virus in the genesis of anterior uveitis is CMV. CMV uveitis is typical of the immunocompromised subject and particularly of patients who have a reduced cellular T response. However, reactivation of CMV uveitis can be observed in originally immunocompetent subjects who are treated with drugs that temporarily depress cellular immunity even locally, such as patients under treatment with corticosteroids and cyclosporine [18]. The prevalence of CMV in particular geographical areas also plays an important role, which explains why this type of uveitis is particularly frequent in Asia [19]. CMV uveitis has the characteristics of unilateral recurrent acute anterior uveitis, but it tends to be chronic in individuals over 40 years of age. It is typical for this uveitis to be associated with increased intraocular pressure that can exceed even 50 mmHg during an acute attack. It is also a characteristic finding that elevated intraocular pressure is not associated with the severity of inflammation. In pediatric patients with herpes simplex virus (HSV) type 1 or type 2 and varicella zoster virus (VZV) infection, uveitis usually presents in an acute unilateral granulomatous or non-granulomatous form [7]. Herpetic anterior uveitis has been described as uncommon in childhood regardless of the geographic region [1] but in more recent literature, its frequency has been reported to be increasing due to improved diagnostic methods on aqueous humor and better clinical knowledge of the disease. Hettinga analyzed 345 children with uveitis and found that the most prevalent viral pathogen was varicella-zoster virus (VZV), which accounted for 39% of cases. 40% of the children underwent aqueous humor analysis that was positive for the presence of the virus in 75% of cases [4]. Rubella virus (RV) uveitis is often unilateral [20] and is a frequent cause of FUS. It should be noted, however, that viral nucleic acids of this virus are rarely detected during uveitis. This finding has led to speculation that in FUS the pathogenetic damage is induced more by the antiviral immune response than by the cytopathic effect of the virus. With the introduction of mandatory anti-rubella vaccination, cases of RV uveitis have been drastically reduced [15]. The anti-rubella inflammatory response appears to be a T-helper 1 (Th1) type response as evidenced by the increased presence of interferon-gamma within the aqueous humor of affected patients [15]. As for posterior viral uveitis, this can be caused by HSV. A distinctive feature of posterior HSV uveitis is the presence of acute retinal necrosis (ARN) or non-necrotizing herpetic retinitis (NNHR). In children, it may manifest as reactivation of asymptomatic neonatal HSV infection [21]. In the absence of ARN, the prognosis is generally good. As for VZV posterior uveitis, this presents with typical progressive external retinal necrosis (PORN) or ARN and is the most frequent cause of atypical necrotizing retinitis [22]. VZV retinitis is, however, rare in children while it typically occurs in elderly individuals possibly preceding the onset of herpes zoster [23]. VZV retinitis lesions generally regress spontaneously or are complicated with severe retinal changes and optic neuropathy [24]. The Epstein–Barr virus, a highly prevalent virus in the population [25], is usually well controlled by the specific T-cell response in healthy subjects. When reactivation occurs, EBV infection can cause at the ocular level a large number of changes including conjunctivitis, episcleritis, keratitis, iritis, optic neuritis, ARN, and retinal vasculitis [26], but tends to cause tissue damage when VZV infection is simultaneously present [27]. The prognosis is generally good, and with adequate antiviral treatment, complete recovery is usually achieved. CMV can also induce posterior-type uveitis [28]. The most frequent ocular manifestations of CMV are retinitis, ARN, and optic neuritis [29,30]. Retinitis can occur in AIDS patients but can also be associated with other conditions of reduced immunocompetence, manifesting in the elderly, in individuals with type two diabetes, or cases of immunodeficiency secondary to the use of cytotoxic or immunosuppressive drugs [30]. CMV retinitis is initially unilateral but then typically extends to the contralateral eye [31] and has a chronic clinical course being asymptomatic in about 50% of cases [32]. Therapy with antiviral drugs in AIDS-infected individuals results in a significantly improved prognosis of retinitis in these subjects [33].

Due to the increased frequency of viral etiology of infectious uveitis, the incidence of toxoplasmosis was low as compared with data from our previous study conducted on patients observed in the period between 1995 and 2004 [6]. It is noteworthy that in that study, toxoplasmosis was the leading cause of pediatric infectious uveitis found in 44% of cases, followed by viral etiology in 17%. According to the present data, however, we observed an inverted ratio between the relative rates of the two infections. Until the 1960s, toxoplasmosis was considered the leading cause of uveitis in children, accounting for 13.5% to 39.4% of cases. Today, in the West, it accounts for 2% to 11% of total cases, while in the East the percentage of toxoplasmosis infection itself remains quite high, having been observed from 7.2% to 25.6% of children with uveitis. These data correlate with the spread of specific parasitosis screening tests and maternal prophylaxis treatments in Europe, which has reduced the frequency and severity of congenital toxoplasmosis [34,35]. Toxoplasmosis is more common in South American children than in European children and presents with more severe ocular pictures. [7]. In our series, Toxocara Canis was confirmed as a rare cause of posterior uveitis (3.5%), in agreement with the findings reported by Hettinga et al. [6]. Fuchs’ heterochromic uveitis was present in 8.7% of cases. It was not possible to confirm the hypothesis suggested by different authors that the disease is strictly associated with RBV infection. Three of the five cases were positive for the IgG Rubella test, one was negative, no serologic tests were performed in one case, and no patients underwent aqueous humor analysis. This is in contrast to what has been reported in other studies [15]. The most frequently observed course was recurrent acute uveitis, present in 47.4% of cases, followed by chronic (38.6%) and acute (14%) uveitis. Our results differ from those of the study by Smith et al. [36], who report a predominance of the chronic course in their case series, probably due to the inclusion of idiopathic uveitis, particularly those associated with juvenile idiopathic arthritis (JIA).

The results of our study show that the visual prognosis was generally good. At the time of diagnosis, visual acuity was >20/30 in 77.8% (44/57) of all children with infectious uveitis and <20/200 in 10.5% (6/57) of cases. Toxoplasma gondii infection was associated with a worse visual prognosis (visual acuity < 20/200) in 57.1% (three pts) of cases, as reported by Smith et al. [36] and by Paroli et al. [6], followed by infections by viral pathogens in 28.6% (two cases) and toxocariasis in 14.3% (one case). In our series, the most frequent complications were cataracts (17.5% of cases), papillitis (21.74% of cases), and posterior synechiae, detected in 19.6% of cases. Rosenberg et al. [37] and De Boer et al. [38] reported a similar frequency of these complications, with cataracts being reported as the leading cause of surgery in pediatric uveitis. Uveitis in children results in significant morbidity due to severe, chronic, and recurrent intraocular inflammation. Any diagnostic delay results in a more severe prognosis than in adults, such as possible visual impairment or blindness [39]. According to Curragh, most patients can maintain good visual acuity with careful ophthalmologic monitoring [40]. Our results suggest that for the etiologic diagnosis of children uveitis possible infectious etiology should always be considered. It is important to remind that Masquerade syndrome, a group of various ocular diseases such as leukemia, lymphoma, neuroblastoma, or intraocular foreign bodies that may mimic uveitis, must be ruled out. Over time, thanks to a multi-disciplinary diagnostic approach, the delay in diagnosis and therapeutic intervention has been greatly reduced, allowing a progressive decrease in idiopathic uveitis and improved visual outcome. Regarding treatment, while early intervention is necessary to preserve vision and prevent the development of further complications, the possible side effects of prolonged steroid therapy and the risk of systemic treatment in a pediatric patient with an immature immune system and an evolving skeletal and reproductive system must be considered.

## 5. Conclusions

Although a reduction in the prevalence of childhood infectious uveitis has been observed over the past decade, it still poses a challenge to the ophthalmologist because of its chronic and asymptomatic course compared with uveitis in adulthood. Proper classification and early diagnosis are necessary for targeted and timely treatment that can lead to a good prognosis in most cases. Our work has some limitations mainly related to the retrospective nature of the study. In addition, it was not always possible to search for pathogen nucleic acids in biological fluids, especially in younger children in whom the diagnosis remains only presumptive. Further studies are needed to evaluate childhood uveitis with newer and minimally invasive diagnostic tools. Finally, we stress the importance of accurate differential diagnosis with noninfectious uveitis because of the risk of inappropriate use of immunosuppressive drugs that can lead to very serious consequences for the vision and health of these young patients.

## Figures and Tables

**Table 1 medicina-58-01673-t001:** Diagnostic tests used in infectious uveitis.

Etiology of Uveitis	Diagnostic Tools
Toxoplasmosis	PCR, IgM, IgG
Streptococcosis	ASO titer
CMV	PCR, IgG
EBV	VCA-IgG, VCA-IgM, EA-IgG, EBNA-IgG
HSV	IgG
VZV	IgG
Toxocariasis	IgG, IgM
Tuberculosis	Quantiferon test, chest X-ray

PCR = Polymerase chain reaction; Ig = immunoglobulin; ASO = anti-streptolysin O.

**Table 2 medicina-58-01673-t002:** Etiology of infectious uveitis.

Type of Infection	N° of Patients	%
Toxoplasmosis	14	24.56
Streptococcosis	2	3.51
Viral infections (total)	32	56.14
CMV	9	15.79
EBV	5	8.77
HSV	11	19.3
VZV	7	12.28
Toxocariasis	2	3.51
Tuberculosis	2	3.51
Fuchs’ heterochromic uveitis	5	8.77

CMV = Cytomegalovirus; EBV = Epstein–Barr virus; HSV = herpes simplex virus; VZV = varicella zoster virus.

**Table 3 medicina-58-01673-t003:** Anatomical location of uveitis and the associated pathogens.

Anatomic Location	N	%	**Etiology**	N	%	*p **
Anterior	21	36.84	Fuchs’	5	23.8	<0.01
			CMV	2	9.5	-
			EBV	1	4.7	-
			HSV	9	42.8	<0.05
			VZV	4	19.04	-
Intermediate	4	7.02	EBV	3	75	-
			CMV	1	25	-
Posterior	23	40.35	CMV	5	21.7	-
			EBV	1	4.3	-
			Post-streptococcal	1	4.3	-
			Toxoplasmosis	14	60.8	<0.01
			Toxocariasis	2	8.6	<0.01
Panuveitis	9	15.79	Post-streptococcal	1	11.1	-
			CMV	1	11.1	-
			HSV	2	22.2	-
			MTB	2	22.2	<0.01
			VZV	3	8.6	-

N = number of patients; * *p* ≤ 0.05 = the anatomic location of uveitis was significantly associated with infection.

**Table 4 medicina-58-01673-t004:** Uveitis characteristics.

Course/Laterality	Anterior	Intermediate	Posterior	Panuveitis
Acute Chronic Recurrent	19.1% 23.8% 57.1%	25% 50% 25%	8.7% 43.5% 47.8%	11.1% 55.6% 33.3%
Unilateral Bilateral	47.37% 52.63%	17.39% 82.61%	67.74% 32.26%	40.00% 60.00%

**Table 5 medicina-58-01673-t005:** Symptoms at the onset of infectious uveitis.

Symptom	N	%
Reduced VA	23	40.3
Hyperemia	21	36.8
Pain	14	24.5
Photophobia	8	14.0
Floaters	10	17.5
Incidental findings	8	14.0

N = number of patients.

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
