# Peer review of "Clinical Features of Infectious Uveitis in Children Referred to a Hospital-Based Eye Clinic in Italy"

_medicina, 2022, doi:10.3390/medicina58111673_

Round 1

Reviewer 1 Report

I thank the editor for allowing me to review this article. 

This article is a descriptive paper describing clinical features of infectious uveitis in  children in a hospital in Italy. 

The authors have lumped most of the text as a single paragraph under each subheading. I would advise the authors to format this accordingly to aid in the readability of the manuscript

I would like to seek clarification from the authors as to how the diagnosis of infectious uveitis was made. Was this predominantly based off a clinical diagnosis? I understand from the text that a range of additional investigations were performed but it is unclear as to whether a diagnosis was made based upon this information. Or based off a clinical diagnosis in pathognomic cases. Information on page 2 will lend itself well perhaps to a table with tests in rows and diagnoses in columns for instance, to provide the readership an understanding of how a clnical diagnosis was made. 

The authors have provided very basic information pertaining to the clinical features of patients presenting with uveitis. There are key pieces of the clinical history and examination findings that are not present. I would have liked for the authors to provide further  information such as the patient's presenting intraocular pressure, whether there were any previous episodes or whether these were fresh cases that have presented. This is important as raised IOP is typically suggestive of a viral aetiology and may prompt further testing. Other key information such as the presence of keratitic precipitates, their size and morphology are all useful information to try to clinch a clinical diagnosis. None of this information was available. 

I would suggest that the authors examine the aforementioned comments and make the appropriate changes to improve on the quality of this manuscript.

Author Response

Query: The authors have lumped most of the text as a single paragraph under each subheading. I would advise the authors to format this accordingly to aid in the readability of the manuscript Answer: The materials and methods and results sections have been divided into subheadings to improve the readability of the text

 Query: I would like to seek clarification from the authors as to how the diagnosis of infectious uveitis was made. Was this predominantly based off a clinical diagnosis? I understand from the text that a range of additional investigations were performed but it is unclear as to whether a diagnosis was made based upon this information. Or based off a clinical diagnosis in pathognomic cases. Answer: The following sentence was added to results: Blood and aqueous humor samples had to be taken for qualitative detection of pathogen DNA by polymerase chain reaction (PCR) in one child with acquired chorioretinal toxoplasmosis and in the aqueous humor of five children with CMV uveitis.

Query: Information on page 2 will lend itself well perhaps to a table with tests in rows and diagnoses in columns for instance, to provide the readership an understanding of how a clinical diagnosis was made. Answer: Table 1 with the required information was added to the manuscript

Query: The authors have provided very basic information pertaining to the clinical features of patients presenting with uveitis. There are key pieces of the clinical history and examination findings that are not present. I would have liked for the authors to provide further information such as the patient's presenting intraocular pressure, whether there were any previous episodes or whether these were fresh cases that have presented. This is important as raised IOP is typically suggestive of a viral etiology and may prompt further testing. Answer: The following sentences were added:  In the other cases a clinical diagnosis was made. Typical fine keratic precipitate was found in anterior viral uveitis, iris atrophy was found in a half of HSV cases and elevated IOP during flare in one eye with HSV and in four eyes with VZV. Toxoplasma gondii was the agent responsible for the majority of infectious uveitis in childhood, being present in 24.5% of cases. A diagnosis of certain congenital toxoplasmosis was possible only in 3 cases through the data of the mother's seroconversion in the second trimester of pregnancy. For the rest, the forms of retinochoroiditis were predominantly unilateral, all involving the posterior pole with compromise of the macula in 6 eyes. The general anamnesis revealed the presence of risk factors in 7 children, in particular 2 ate raw meat and 5 had a kitten with which they had regular contact.

Query: Other key information such as the presence of keratic precipitates, their size and morphology are all useful information to try to clinch a clinical diagnosis. None of this information was available. Answer:  The following sentence was added in results. Typical fine keratic precipitate was found in anterior viral uveitis, iris atrophy was found in a half of HSV cases and elevated IOP during flare in one eye with HSV and in four eyes with VZV.

Reviewer 2 Report

This article has presented epidemiological data on the incidence of uveitis in the paediatric population at the authors' centre, with a particular focus on uveitis with an infectious background. The assumptions of the paper seem interesting considering the timely implementation of appropriate treatment as well as the prevention of complications.

Among the references, those published in recent years including 2021 were missing; the paper is epidemiological and requires up-to-date published data, which is in accordance with the follow-up in the current study. No merit in citing epidemiological data prior to 2009.

The introduction has been very poor. Suggestions include adding information on what the most common uveitis-causing infections have been mentioned in the literature in relation to geographical regions. What was the impact on the course of uveitis, treatment, complications of each infection according to clinical knowledge and literature data. The range from 6 to 33% was very large from a statistical perspective and requires the detail to be added to allow the reader to recognise in which countries or for what reasons some studies report a prevalence of less than 10% and others reaching up to a third of patients. It has not been clear from the introduction why identification of infection is important.

The methods described numerous tests, including laboratory tests, which were performed on patients and which have not been mentioned by the authors in the results or discussion.

The mean of the follow-up time should be added to the results section and replaced by the median (similarly, the median would be suggested for mean age).

The sentence probably requires a linguistic correction: Toxoplasma gondii was the agent responsible for the majority of infectious uveitis in childhood, being present in 24.5% of cases. Viral infections were responsible for 56.14% of the cases when considered as a whole. Similar to the sentence: Streptococcal and Mycobacterium tuberculosis infection caused uveitis in 3.5% of cases.

The results should be systematised, made more readable. In Table 2, the % could be shown in the last column. Perhaps some of the results could be presented in table form only.

It has not been clear from the results how many patients and for what reason had visual impairment/decreased visual acuity after the follow-up period. Did cataracts occur in 57.14% of patients with reduced visual acuity or in 57.14% of all patients? The methods should indicate what is meant by visual acuity >20/30. 

It would seem interesting to know if any symptoms were particularly more common with a particular type of infection or location of uveitis - such data were missing in the results.

The discussion has a missing reference in the first sentence. The age of the patients to be part of the results. The discussion has not discussed in the order of the results presented - the results started with a description of the frequency of the different infectious agents and the discussion started with the course of the infection. 

Some sentences in the discussion seem to diverge from both the main topic and the paediatric patients.

Statistics could be performed for the data presented in the tables (multivariate tables).

It was not specified when the end of patient follow-up occurred.

The discussion lacks consideration of whether the prevalence of a particular infections agent depends on the geographical region, etc.

Conclusion: It is not clear from the study which symptoms were more frequent with infectious agents and which pathogen. It has not been shown how the treatment affected the preservation of visual acuity.

Author Response

Query: Among the references, those published in recent years including 2021 were missing; the paper is epidemiological and requires up-to-date published data, which is in accordance with the follow-up in the current study. No merit in citing epidemiological data prior to 2009.Answer: An up-to-date reference has been added: Pediatric uveitis: A comprehensive review. Maleki A, et al. Surv Ophthalmol. 2022. PMID: 34181974 Review. References citing epidemiological data prior to 2009 have been deleted. Now total references are 20 less than previous, which in our opinion is also more in line with a research article.

Query: The introduction has been very poor. Suggestions include adding information on what the most common uveitis-causing infections have been mentioned in the literature in relation to geographical regions. What was the impact on the course of uveitis, treatment, complications of each infection according to clinical knowledge and literature data. The range from 6 to 33% was very large from a statistical perspective and requires the detail to be added to allow the reader to recognise in which countries or for what reasons some studies report a prevalence of less than 10% and others reaching up to a third of patients. It has not been clear from the introduction why identification of infection is important. Answer: The following sentences here in bold have been added to the introduction: Pediatric uveitis account for 5-10% of the total uveitis. They lead to significant morbidity that increases  with the duration of the disease and since the diagnosis is delayed, the prognosis is worse when compared with most of the adult cases (( Ilknur Tugal-Tutkun  :  Pediatric uveitis J Ophthalmic Vis Res . 2011 Oct;6(4):259-69.). Most studies on uveitis in children have focused on the non-infectious form. However, infectious uveitis often has a poor prognosis because of its high tendency to become chronic [1, 2]. The most frequent ocular complications consist of the development of cataracts, band keratopathy, glaucoma, amblyopia, and macular disease [3]. All of these conditions can lead to reduced visual acuity and even blindness. In many studies, an infectious cause of uveitis has been identified in 6-33% of children seen in tertiary referral centers [4-10]. As in adults, the prevalence of infectious uveitis varies from country to country in conjunction with new migratory flows and particular socio-economic conditions of the population. It is above all the populations of the Middle East, Turkey and Africa that have most of the infectious cases.( ( Ilknur Tugal-Tutkun  :  Pediatric uveitis J Ophthalmic Vis Res . 2011 Oct;6(4):259-69. )  In developing countries tuberculosis is a significant cause of childhood uveitis while post-streptococcal uveitis was observed mainly in in Europe. (Pediatric uveitis: A comprehensive review. Maleki A, et al. Surv Ophthalmol. 2022. PMID: 34181974 Review.) In Italy an infectious origin has been detected one third of uveitis cases in pediatric population (9). Regarding the prevalence of different etiologies, toxoplasmosis is reported to be the most frequent cause of infectious uveitis worldwide, followed by viral forms and Toxocara canis. (6)The difficulty and delay in diagnosis, limited treatment options, and the effect of the disease throughout adult life are major challenges for pediatric ophthalmologists. Therefore, the ability to identify infectious uveitis early is extremely important, as the treatment regimens differ significantly between infectious and non-infectious uveitis. Purpose of this retrospective cohort study was to analyze the etiology, clinical features, ocular complications, and visual prognosis of a series of children with infectious uveitis referred to our tertiary eye care center and to compare the results obtained with those available in the literature.

Query: The methods described numerous tests, including laboratory tests, which were performed on patients and which have not been mentioned by the authors in the results or discussion. Answer: The sentence: Blood and aqueous humor samples for the qualitative detection of pathogen DNA by polymerase chain reaction (PCR) was necessary in one child with chorioretinal toxoplasmosis and in aqueous humor in two children with CMV anterior uveitis has been added in the results

Query: The mean of the follow-up time should be added to the results section and replaced by the median (similarly, the median would be suggested for mean age). Answer: The sentences were added to results: Mean values have been changed to median values

Query: The sentence probably requires a linguistic correction: Toxoplasma gondii was the agent responsible for the majority of infectious uveitis in childhood, being present in 24.5% of cases. Viral infections were responsible for 56.14% of the cases when considered as a whole. Similar to the sentence: Streptococcal and Mycobacterium tuberculosis infection caused uveitis in 3.5% of cases. Answer: The sentences have been modified eliminating “for the majority of infectious uveitis in childhood, being present”. The term “both” was added to the sentence regarding Streptococcal and Mycobacterium tuberculosis infection

Query: The results should be systematised, made more readable. In Table 2, the % could be shown in the last column. Perhaps some of the results could be presented in table form only. Answer: The sentence and Table 1 summarizes the infectious etiologies of children's uveitis as found in this study was deleted. The references to tab.1 and tab.2 have been added to the results (in red). Tab 2 has been modified

Query: It has not been clear from the results how many patients and for what reason had visual impairment/decreased visual acuity after the follow-up period. Did cataracts occur in 57.14% of patients with reduced visual acuity or in 57.14% of all patients? The methods should indicate what is meant by visual acuity >20/30. Answer: The following sentence was added to results: Cataract occurs in 57.14% of the eyes with visual acuity <20/30. The sentences: a best corrected visual acuity (BCVA) using Snellen charts or illiterate E charts and BCVA was measured both in foots and in Logmar. Visual acuity loss was defined as crossing 20/50 or worse (visual impairment) and 20/200 or worse (legal blindness) was added to Material and Methods.

Query: It would seem interesting to know if any symptoms were particularly more common with a particular type of infection or location of uveitis - such data were missing in the results. Answer: The following sentence was added to results: A reduced visual acuity was found in all the patients with posterior uveitis irrespective of the etiology while viral anterior uveitis was related to hyperemia and photophobia. An incidental finding was mainly due to intermediate viral uveitis.

Query: The discussion has a missing reference in the first sentence. The age of the patients to be part of the results. The discussion has not discussed in the order of the results presented - the results started with a description of the frequency of the different infectious agents and the discussion started with the course of the infection. Answer: The following reference was added : Ilknur Tugal-Tutkun  :  Pediatric uveitis J Ophthalmic Vis Res . 2011 Oct;6(4):259-69.; The sentences: The most frequently observed course was recurrent acute uveitis present in 47.4% of cases, followed by chronic (38.6%) and acute (14%) uveitis. Our results differ from those of the study by Smith et al. [12], reporting that chronic course predominates, probably because of the inclusion in their case series of idiopathic uveitis, and in particular those associated with juvenile idiopathic arthritis (JIA)It has moved to the bottom of the discussion

Query: Some sentences in the discussion seem to diverge from both the main topic and the paediatric patients. Answer: the sentence “Panuveitis was present in 16% of cases, of which 66.7% were related to viral agents, 22.2% to Mycobacterium tuberculosis infection, and 11.1% to toxoplasmosis was deleted because mistakenly repeated two times. The sentence: In pediatric patients Tipe1 and 2 herpes simplex virus (HSV) and varicella zoster virus (VZV) can present as acute unilateral granulomatous or non-granulomatous uveitis ((Pediatric uveitis: A comprehensive review. Maleki A, et al. Surv Ophthalmol. 2022. PMID: 34181974 Review.) Herpetic anterior uveitis was described as uncommon in childhood irrespective of geographic region (Ilknur Tugal-Tutkun  :  Pediatric uveitis J Ophthalmic Vis Res . 2011 Oct;6(4):259-69.) but the frequency has increased in the most recent literature thanks to diagnostic methods on aqueous humor and better clinical knowledge of the disease. Hettinga analyzed 345 children with uveitis and found that the most prevalent viral pathogen was varicella-zoster virus (VZV), representing 39% of cases. Many children 40% were submitted to aqueous humor analysis that was positive 75% of cases” was added in discussion

Query: Statistics could be performed for the data presented in the tables (multivariate tables). Answer: Multivariate analysis was performed were possible and results reported in the result section.

Query: It was not specified when the end of patient follow-up occurred. Answer: median follow-up was 23 months.

Query: The discussion lacks consideration of whether the prevalence of a particular infectious agent depends on the geographical region, etc. Answer: The sentence Herpetic anterior uveitis was described as uncommon in childhood irrespective of geographic region was added. The sentence The incidence is higher in South American than in European children showing more severe ocular picture was added to discussion

Query: Conclusion: It is not clear from the study which symptoms were more frequent with infectious agents and which pathogen. It has not been shown how the treatment affected the preservation of visual acuity. Answer: The sentence: a reduced visual acuity was found in all the patients with posterior uveitis irrespective of the etiology while viral anterior uveitis was related to hyperemia and photophobia. An incidental finding was mainly due to intermediate viral uveitis was added in results. The sentence: in fact, a targeted and timely treatment in our patients has allowed a good prognosis in 75% of cases has been added

Round 2

Reviewer 1 Report

The authors report that most cases of infectious uveitis involves Middle Eastern, Turkish and African populations, before discussing tuberculosis as a significant cause of childhood uveitis in developing countries - which is it exactly? 

Page 2 - BCVA has an additional V included

The authors describe measurement of BCVA in feet and LogMAR (please note its classically described as LogMAR rather than logmar).

Page 3 - 10.6 rather than 10,6 years. Please be sure to report IQR for median values and SD for means. 

The 24.5 percent of cases - “the should be dropped”, percentage written as %

For the remaining cases, the forms of retinochoroiditis were predominantly unilateral, all involving the posterior pole with macula impairment in 6 eyes - do the authors mean that toxoplasmosis was diagnosed clinically based on the presence of a chorioretinal lesion? 

Kitten is specific to a small cat. Cats may be the more appropriate term the authors might want to consider using. 

How were patients with TB associated uveitis diagnosed? 

“The most frequent cause of visual acuity reduction at the end of the follow-up period was total cataract present” - what does total cataract present mean? 

Cataracts occurred in 10 patients and there were 57 patients recruited. This does not comprise 21.7% of cases. Would suggest the authors relook their numbers or method of expression 

I find it interesting that the authors have chosen 20/30 as a cut-off for visual acuity. Why is this so? The WHO definition of vision impairment -  which is typically used globally, uses VA worse than 20/40 as a cut-off. This is also a standard used globally in many countries for driving etc. 

The authors have added “reduced visual acuity in all patients with posterior uveitis”. What is the definition of reduced visual acuity - is this based off a line or more of vision loss?

Author Response

Referee #1 (2 Roundup)

Query: The authors report that most cases of infectious uveitis involve Middle Eastern, Turkish and African populations, before discussing tuberculosis as a significant cause of childhood uveitis in developing countries - which is it exactly? Answer: The statement has been modified in the discussion section: “As in adults, the prevalence of infectious uveitis varies from country to country depending on migration flows and particular socioeconomic conditions. Most cases of infectious uveitis involve populations in the Middle East, Turkey, and Africa. In Italy, an infectious origin has been found in one third of cases of uveitis in the pediatric population. Regarding the prevalence of the different etiologies, toxoplasmosis appears to be the most frequent cause of infectious uveitis worldwide, followed by forms with viral etiology and Toxocara canis. In developing countries, tuberculosis is a significant cause of childhood uveitis, while post-streptococcal uveitis is mainly reported in Europe.

Query: Page 2 - BCVA has an additional V included

The authors describe measurement of BCVA in feet and LogMAR (please note its classically described as LogMAR rather than logmar). Answer: The spelling error has been corrected. It was specified that BCVA was measured in feet.

Query: Page 3 - 10.6 rather than 10,6 years. Please be sure to report IQR for median values and SD for means. Answer: Numbers comma has been changed to dot. IQR has been added and expressed as Q1-Q3 range.

Query: The 24.5 percent of cases - “the should be dropped”, percentage written as % Answer: The term "percent" has been changed to "%" throughout the text.

.

 Query: For the remaining cases, the forms of retinochoroiditis were predominantly unilateral, all involving the posterior pole with macula impairment in 6 eyes - do the authors mean that toxoplasmosis was diagnosed clinically based on the presence of a chorioretinal lesion? Answer: The finding of focal necrotizing retinochoroiditis as a typical clinical sign of ocular toxoplasmosis was specified in the text.

Query: Kitten is specific to a small cat. Cats may be the more appropriate term the authors might want to consider using. Answer: Kitten has been changed to cat

Query: How were patients with TB associated uveitis diagnosed? Answer: Quantiferon test and chest X-ray were performed for TB diagnosis. Table 1 has been modified accordingly.

Query: “The most frequent cause of visual acuity reduction at the end of the follow-up period was total cataract present” - what does total cataract present mean? Answer: A comma was added after the word cataract to clarify the meaning of the sentence.

Query: Cataracts occurred in 10 patients and there were 57 patients recruited. This does not comprise 21.7% of cases. Would suggest the authors relook their numbers or method of expression Answer: Data have been corrected. Cataract cases were expressed either as % of cases or % of eyes.

Query:I find it interesting that the authors have chosen 20/30 as a cut-off for visual acuity. Why is this so? The WHO definition of vision impairment - which is typically used globally, uses VA worse than 20/40 as a cut-off. This is also a standard used globally in many countries for driving etc. Answer: 20/30 has been changed to 20/50

Query: The authors have added “reduced visual acuity in all patients with posterior uveitis”. What is the definition of reduced visual acuity - is this based off a line or more of vision loss? Answer: The meaning of reduced visual loss has been specified (BCVA with loss of vision of one or more lines in Snellen tables or illiterate E tables)

Reviewer 2 Report

The submitted version of the manuscript includes the previously proposed improvements. Minor corrections are suggested below.

Necessary improvements:

I have not found any results to those tests described in the research methods in the results and discussion section.: “Laboratory investigations included common blood tests such as complete blood count, analysis of inflammation proteins such as ESR and CRP, protidogram, proteinemia, and serum immunoglobulin level. Laboratory investigations also included HLA typing and immunological tests, such as the detection of antinuclear antibodies (ANA), antiphospholipid antibodies, anti-neutrophil cytoplasmic antibodies (cANCA/pANCA), and angiotensin-converting enzyme test (ACE-test).” Perhaps a "complete blood count, analysis of inflammatory proteins such as ESR and CRP, protidogram, proteinemia and serum immunoglobulin levels" would be helpful in early detection of the infectious cause of uveitis? If they are not mentioned in the results they should be removed from the methods.

This is group characteristic: “The diagnosis of infectious uveitis was made in 57 out of 314 cases of pediatric uveitis due to all causes (18.1%). Thirty children were female (52.6%) and 27 were male (47.4%), the median age at diagnosis was 10,6 years. Median follow-up was 23 months.”

This sentence should not be part of the aetiology but of the complications.: “Typical fine keratic deposits were found in all viral anterior uveitis, iris atrophy was found in half of HSV cases, and elevated IOP was detected during an Elevated IOP was detected during a flare episode in one eye with HSV and in four eyes with VZV.”

In the methods visual acuity has a classification of > 20/200 20/50 and in the results section 20/30 is used. For non-specialists in ophthalmology this is unclear.

Multivariate regression analysis is barely described and if it is not statistically significant it does not seem to need to be added.

When reporting the median, the IQR should be given.

"These findings are in with those reported by Hetting et al." It is suggested to describe these results. Over 18% seems quite a large number, what prevalence is reported in other studies that use the phrase "rare" and what prevalence is in adults?

In the reviewer's opinion, insufficient attention is paid to clinical features and basic laboratory investigations as per the posited summary "This study highlights that increased knowledge of the different clinical features and conditions associated with paediatric infectious uveitis along with the importance of close collaboration with the physician. and conditions associated with infectious uveitis in children, along with the importance of close collaboration with specialists from other disciplines can lead to timely diagnosis and followed by more effective treatment, which can prevent visual deterioration and further complications. complications." It is undoubtedly clear from the study that infectious causes should be kept in mind and diagnosed/excluded, the authors show that they are not all that rare. But it is not clear which clinical features to look out for, which is statistically significantly more frequent for a given infection.

Suggested amendments

Please use statistical tests to answer whether, for example, posterior uveitis was significantly more frequent with Toxpolasma infection. Similarly, it is suggested to perform relevant statistics showing the significance of the association of the site of inflammation, gender, age, etc. with the site of inflammation, bilateral, clinical course.... Was treatment time longer for any particular infection? Was the deterioration of vision statistically more significant with any one infectious agent?

Multivariate regression analysis is barely described and if it is not statistically significant it does not seem to need to be added. Perhaps more variables should be added to this analysis? It is suggested (as above) to add less advanced statistical tests e.g. chi2 in order to search for significant correlations.

Author Response

Referee #2 (2 Roundup)

The submitted version of the manuscript includes the previously proposed improvements. Minor corrections are suggested below.

Necessary improvements:

Query: I have not found any results to those tests described in the research methods in the results and discussion section.: “Laboratory investigations included common blood tests such as complete blood count, analysis of inflammation proteins such as ESR and CRP, proteinogram, proteinemia, and serum immunoglobulin level. Laboratory investigations also included HLA typing and immunological tests, such as the detection of antinuclear antibodies (ANA), antiphospholipid antibodies, anti-neutrophil cytoplasmic antibodies (cANCA/pANCA), and angiotensin-converting enzyme test (ACE-test).” Perhaps a "complete blood count, analysis of inflammatory proteins such as ESR and CRP, proteinogram, proteinemia and serum immunoglobulin levels" would be helpful in early detection of the infectious cause of uveitis? If they are not mentioned in the results they should be removed from the methods. Answer: The section on diagnostic methods has been removed from the methods section, as suggested.

Query: This is group characteristic: “The diagnosis of infectious uveitis was made in 57 out of 314 cases of pediatric uveitis due to all causes (18.1%). Thirty children were female (52.6%) and 27 were male (47.4%), the median age at diagnosis was 10,6 years. Median follow-up was 23 months.” Answer: These sentences are now in the group characteristics. Median values have been expressed together the interquartile range (Q1-Q3)

Query: This sentence should not be part of the aetiology but of the complications.: “Typical fine keratic deposits were found in all viral anterior uveitis, iris atrophy was found in half of HSV cases, and elevated IOP was detected during an Elevated IOP was detected during a flare episode in one eye with HSV and in four eyes with VZV.” Answer: The sentence has been moved in the “Ocular complication” section.

Query: In the methods visual acuity has a classification of > 20/200 20/50 and in the results section 20/30 is used. For non-specialists in ophthalmology this is unclear. Answer: 20/30 has been corrected to 20/50.

Query: Multivariate regression analysis is barely described and if it is not statistically significant it does not seem to need to be added. Answer: Multivariate regression analysis has been deleted from the “Statistical analysis” section. The sentences “Chi-square (χ²) test was used for categorical data. Differences between groups were considered significant when the P value was < 0.05. Numerical values were presented as percentages or as medians and interquartile ranges (Q1-Q3)” have been added.

Query: When reporting the median, the IQR should be given. Answer: The interquartile range was added in parentheses after the median values and was expressed as Q1-Q3 values.

Query: "These findings are in with those reported by Hetting et al." It is suggested to describe these results. Over 18% seems quite a large number, what prevalence is reported in other studies that use the phrase "rare" and what prevalence is in adults? Answer: The statement: Infectious uveitis in pediatric age is a rare disease compared with adults. In a recent Italian study on uveitis of all causes, it was reported that 30% of cases were infectious in origin, but only 12% of these were in childhood. The results obtained from the present study of a cohort of Italian children showed that 18.5% of uveitis from all causes was infectious in origin and illustrate a decrease in prevalence compared to 31% as reported in a previous study in the same tertiary referral center” was added. The sentence citing the study by Hetting et al. has been corrected.

Query: In the reviewer's opinion, insufficient attention is paid to clinical features and basic laboratory investigations as per the posited summary "This study highlights that increased knowledge of the different clinical features and conditions associated with paediatric infectious uveitis along with the importance of close collaboration with the physician. and conditions associated with infectious uveitis in children, along with the importance of close collaboration with specialists from other disciplines can lead to timely diagnosis and followed by more effective treatment, which can prevent visual deterioration and further complications. complications." It is undoubtedly clear from the study that infectious causes should be kept in mind and diagnosed/excluded, the authors show that they are not all that rare. But it is not clear which clinical features to look out for, which is statistically significantly more frequent for a given infection. Answer: The discussion section has been modified accordingly. Table 1 shows the diagnostic tests to be performed for differential diagnosis, and Table 3 shows the frequencies of anatomical location of uveitis for each infectious agent. 

Suggested amendments

Query: Please use statistical tests to answer whether, for example, posterior uveitis was significantly more frequent with Toxpolasma infection. Similarly, it is suggested to perform relevant statistics showing the significance of the association of the site of inflammation, gender, age, etc. with the site of inflammation, bilateral, clinical course.... Was treatment time longer for any particular infection? Was the deterioration of vision statistically more significant with any one infectious agent? Answer: Chi-square test was used for comparison. Results and resulting  P values have been added to Table 3.

Query: Multivariate regression analysis is barely described and if it is not statistically significant it does not seem to need to be added. Perhaps more variables should be added to this analysis? It is suggested (as above) to add less advanced statistical tests e.g. chi2 in order to search for significant correlations. Answer: Chi-square test analysis has been performed instead of multivariate regression analysis